# Effects of Juvenile Hormone Analog and Days after Emergence on the Reproduction of Oriental Armyworm, *Mythimna separata* (Lepidoptera: Noctuidae) Populations

**DOI:** 10.3390/insects13060506

**Published:** 2022-05-27

**Authors:** Weixiang Lv, Liting Zeng, Zhe Zhang, Hengguo He, Fang Wang, Xingcheng Xie

**Affiliations:** 1Key Laboratory of Southwest China Wildlife Resources Conservation, China West Normal University, Nanchong 637002, China; lvwx@cwnu.edu.cn (W.L.); zenglt2002@163.com (L.Z.); hengguohe@163.com (H.H.); 2School of Architecture and Engineering, Zhengzhou Business University, Gongyi 451200, China; Zc67220055@163.com; 3State Key Laboratory for Biology of Plant Diseases and Insect Pests, Institute of Plant Protection, Chinese Academy of Agricultural Sciences, Beijing 100193, China

**Keywords:** *Mythimna separata*, migrant, resident, juvenile hormone analog, sensitive stage, reproduction

## Abstract

**Simple Summary:**

Flight and reproduction are two major life history traits for coping with changing environments in migratory insects. The phenomenon of oogenesis-flight syndrome (namely, the trade-off between migration and reproduction) is regulated by juvenile hormone (JH). The oriental armyworm, *Mythimna separata* (Walker), is an important migratory agricultural pest with strong reproductive capacity. Previous studies have focused on discussions about the effects of JH on *M**. separata* migrants, but little has been known about the potential influences on the residents until now. In this study, the effects of juvenile hormone treatment and age (namely, days after adult emergence) on both migrants and residents of *M. separata* have been studied. Our results showed that the effects of JH analog (JHA) treatment on reproduction depended on adult age of exposure to JHA and populations. The first two days and only the first day after adult emergence were the sensitive period for the exposure of residents and migrants to JHA on ovarian and reproductive development, respectively.

**Abstract:**

*Mythimna separata* (Walker) is a main cereal crop pest that causes extensive damage to the world grain production. The effects of juvenile hormone on *M. separata* populations remain poorly understood. Here, we explored how JH analog (JHA) affected reproductive traits of both migrant and resident populations in this pest. Our results showed that the influence of JHA treatment on reproduction depended on adult age (days after emergence) of exposure to JHA and populations. Exposure of *M. separata* residents to JHA (methoprene) on day 1 and day 2 after adult emergence significantly shortened the pre-oviposition period, but increased the lifetime fecundity, mating frequency and grade of ovarian development compared to the controls. However, *M. separata* migrants exposed to JHA only on day 1 facilitated their reproduction, resulting in a reduction in the pre-oviposition period but an increase in lifetime fecundity, mating frequency and grade of ovarian development. In addition, exposure to JHA from day 2 to day 4 did not significantly affect the ovarian and reproductive development in both migrant and resident populations. These results indicated that the first two days after adult emergence were the sensitive period for residents. In contrast, only the first day after adult emergence was the sensitive stage for migrants. Our findings will contribute to a better understanding of JHA function on *M. separata* populations.

## 1. Introduction

Juvenile hormone (JH) plays critical roles in development, metamorphosis, reproduction, migration and pheromone production in insects [1,2,3]. In migratory insects, the onset of migratory behaviour usually occurs in the pre-reproduction period, which is initiated by sexually immature adults. There thus exists a trade-off between migration and reproduction, and the phenomenon of oogenesis-flight syndrome is regulated by JH [4,5,6]. Until recently, some studies in regard to *Melanoplus sanguinipes* (F.), *Cnaphalocrocis medinalis* (Guenee) and other insects have demonstrated that JH levels are closely related to the development of ovaries and flight muscles. Higher JH levels accelerate ovarian development, resulting in the degradation of flight muscles and inhibition of migratory ability. However, the absence or a low concentration of JH inhibits the development and fecundity of the ovaries, stimulating migratory behavior in adults [7,8]. Moreover, research on honeybees, *Apis mellifera,* found that the lifespan of adults was associated with the JH levels [9].

Insect migration and outbreaks have a close association with diverse adaptive strategies that enable relocation from deteriorating habitats to suitable breeding habitats [10]. Previous study revealed that external environmental factors as well as the days after emergence could influence the migratory strategies in *Spodoptera exigua* (Hübner) [11] and *Loxostege sticticalis* L. [12]. Recent works have showed that a short critical period is involved in the regulation roles of migration and reproduction in most migratory insects [13,14,15]. During this critical period, extreme environmental conditions such as starvation, a long photoperiod and a low temperature can up-regulate the expression of the *allatotropin* (AT) gene and accelerate the synthesis of JH by the corpora allata (CA), which thereby may lead to the increased level of JH and result in the early spawning and decreased flight ability in female moths [16]. In the beet webworm, starvation in the sensitive period, which is the first 2 days after adult emergence, may reverse the decision of migration in the immature stage but contribute to the adult reproduction by compressing the development of migratory flight muscle [17]. In contrast with the research on the migrants, however, very limited attention has been given to the residents of migratory insects [18].

Many migratory species have evolved migrant and resident dimorphisms to adapt to different environmental conditions. Examples include alate and apterous morphs of the aphids [19], long winged and wingless crickets [20] and gregarious and solitary locusts [21], where migrants show poorer reproductive capacity but stronger flight ability than residents. Unlike exopterygote insects, endopterygote insects, including lepidopterans, cannot be morphologically identified from migrants and residents alone but can be distinguished by the length of the pre-oviposition period (POP), flight capacity and fecundity. Migrants usually have a longer pre-oviposition period, stronger flight capacity and lower fecundity than residents [18,22]. The length of the pre-oviposition period is an important indicator of the migratory period and propensity and is thus regarded as a key criterion for distinguishing between residents and migrants in most migratory species [17].

*Mythimna separata* (Walker) (Lepidoptera: Noctuidae) is one of the most destructive agricultural pests in both Asia and Oceania, with annual outbreaks in nearly 30 countries [23,24]. The first takeoff of migratory *M. separata* individuals occurs in the first two days after adult emergence [25], and the level of JH and grade of ovarian development in the immigrant population exceed that of the emigrant population after long-distance migration [26]. Levels of JH in the residents were significantly greater than those in the migrants on the first to sixth day after emergence [27]. Jiang and Luo also suggested that lower JH levels favored adult migration, whereas higher levels stimulated oogenesis [28]. Luo et al. [29] demonstrated that the effect of JHA on flight capacity and energy expenditure of *M. separata* migrants differed with the days after adult emergence. During this sensitive period of the first day after adult emergence, JH may regulate the shift from migrants to residents in *M. separata* [27,30]. Our previous study further found that *M. separata* residents exhibited slower development, lower triglyceride content and weaker flight capacity but greater reproductive capacity than the migrants [18]. This study raises important questions as follows. (i) Do JHA and age (the days after adult emergence) have effects on the reproduction of *M. separata* residents? (ii) Is the sensitive period of *M. separata* residents and migrants the same? To better address these questions above, we investigated the effects of *M. separata* residents and migrants exposed to JHA at various ages on their reproduction in the laboratory.

## 2. Materials and Methods

### 2.1. Insect Rearing

The oriental armyworms used in this experiment were from migrant and resident populations maintained in the Key Laboratory of Southwest China Wildlife Resources Conservation, China West Normal University, which were originally collected from field pupae in the Nanning area of Guangxi Province, China. Migrants were reared in round glass jars (9 cm × 13 cm, diameter × height) at a density of 10 larvae per container for at least ten generations before the experiment [29]. Residents were cultured alone in round glass jars (9 cm × 13 cm, diameter × height) for at least eight generations [18]. To keep the population stability of migrants and residents, the rearing methods were continuously adopted according to the larval density of 10 and 1 larvae per container, respectively. All tested larvae were reared with fresh corn seedlings (*Zea mays*) until pupae. After adult emergence, the adults from migrant and resident populations were individually paired (one female and one male) in cylindrical plastic cages (10 cm × 20 cm, diameter × height), provided with 10% honey solution (*v/v*) as the food source until death. All tested individuals from the stage of eggs to adults were maintained under a 14 h light: 10 h dark photoperiod regime at 25 °C with a 70% relative humidity.

### 2.2. JHA Treatment

After adult emergence, newly emerged moths from migrant and resident populations were provided with a juvenile hormone analogue, methoprene (Sigma Chemical Co., Louis, MI, USA), according to the techniques described by Zhang et al. [27]. Our previous study found that the application of 6 μg/μL JHA significantly accelerates adult reproduction and suppresses flight capacity [27]. Hence, we chose the 6 μg/μL JHA as the test concentration to study JH function in this experiment. Briefly, either 5 μL JHA (6 μg/μL) or acetone solution (control) was dropped onto the clean pronotum with a pipette at 15:00 p.m. every day. Subsequently, one female and one male from the same treatment group were individually paired and reared in a cylindrical plastic cage (10 cm × 20 cm, diameter × height) with 10% honey solution (*v/v*). All tested adults were kept under the same laboratory conditions as mentioned above. Either JHA or control treatment was replicated more than 26 times.

### 2.3. Adult Age Treatment

Adult age (days after emergence) in this experiment refers to the days after emergence. The first day of adult emergence is considered day 1, the next day is day 2, and so forth. The moths from migrant and resident populations were exposed to JHA on day 1, day 2, day 3 and day 4 after adult eclosion. The moths exposed to acetone at the same age between these two populations served as controls. In this study, all female and male oriental armyworms from two populations at the same treatment were individually paired before the JHA or acetone exposure. Each age treatment from the two populations was replicated more than 26 pairs.

### 2.4. Reproductive Parameter Determination

To investigate the effects of JHA and age (days after emergence) after adult emergence on the reproduction of *M. separata* migrants and residents, the following reproductive parameters (namely, the pre-oviposition period, lifetime fecundity, oviposition period, adult longevity, mating frequency, mating percentage and the grade of ovarian development) were measured for both JHA and control treatments between migrants and residents, following the methods described in our previous studies [18,31]. The pre-oviposition period refers to the duration from emergence to the first oviposition of adults. The oviposition period was the total number of days between the first and last oviposition. Lifetime fecundity refers to the total number of eggs laid by per female. After female death, mating frequency was computed by counting the number of spermatophores in each spermatheca. Females with spermatophores indicated the success of copulation, whereas those females without spermatophores represented the failure of copulation. The mating percentage was determined by the percentage of females with spermatophores among the total treated females. Ovary data were recorded after female death to determine the grade of ovarian development. The calculation of the ovarian development levels followed the five-level standard, which had been described by Chen et al. [32].

### 2.5. Data Analysis

All data from the experiments were presented as means ± standard errors (SEs). All observed data in this study were tested for a normal distribution by the Shapiro–Wilk test before statistical analysis. All reproductive parameters were analyzed by three-way analysis of variance (ANOVA), with population (Residents or Migrants), JHA treatment (JHA or acetone) and exposure age after adult emergence (including 1, 2, 3 and 4 days old) as factors, followed by Tukey’s honestly significant difference (HSD) test (*p* < 0.05) for mean comparisons. Mating percentage data were arcsine transformed. Pearson’s correlation analysis was applied to investigate the effects of JHA and age on reproductive performance of *M. separata* populations. All statistical analyses were performed with SPSS software (version 22.0; SPSS).

## 3. Results

### 3.1. Reproductive Performance of M. separata between Residents and Migrants Exposed to JHA at Various Ages

Populations significantly affected the pre-oviposition period (POP) and lifetime fecundity (Table 1). The POP of residents on day 1 to day 4 was significantly lower than that of migrants treated with JH analog (JHA) (Day 1: *F*_1.58_ = 19.80, *p* < 0.001; Day 2: *F*_1.59_ = 19.70, *p* < 0.001; Day 3: *F*_1.54_ = 3.84, *p* = 0.050; Day 4: *F*_1.57_ = 5.15, *p* = 0.027, Figure 1A) and acetone (Day 1: *F*_1.58_ = 7.18, *p* = 0.010; Day 2: *F*_1.56_ = 6.07, *p* = 0.017; Day 3: *F*_1.57_ = 5.07, *p* = 0.028; Day 4: *F*_1.58_ = 5.76, *p* = 0.020, Figure 1B). Exposure of residents on day 1 and day 2 to JHA had significantly greater lifetime fecundity compared to the migrants in the JHA treatments (Day 1: *F*_1.58_ = 4.87, *p* = 0.031; Day 2: *F*_1.59_ = 6.77, *p* = 0.012, Figure 1C), but there was no significant difference in lifetime fecundity between residents and migrants in the control treatments (Figure 1D). Similarly, no significant differences in oviposition period (Figure 1E–F), mating frequency (Figure 1G–H) or mating percentage (Figure 2A,B) were observed between residents and migrants exposed to JHA or acetone.

Significant effects of population on female longevity were observed (Table 1), confirming the shorter female longevity of residents exposed to JHA compared to that of migrants on day 1 (*F*_1.58_ = 7.45, *p* = 0.008) and day 2 (*F*_1.59_ = 5.29, *p* = 0.025, Figure 2C). However, no significant differences in female longevity between residents and migrants in the control treatments were found (Figure 2D). There were also no significant differences in male longevity (Figure 2E,F) or ovarian development grade (Figure 2G,H) between residents and migrants in both JHA and control treatments. The correlation analysis suggested that population was negatively correlated with the reproduction of *M. separata* (Table 2).

### 3.2. Effect of JHA on Reproductive Performance of M. separata Populations at Various Ages

JHA treatment and the interaction between JHA treatment and age significantly affected the POP (Table 1). In residents, the POP of resident moths exposed to JHA was significantly decreased compared to the control group on day 1 (*F*_1.58_ = 41.63, *p* < 0.001) and day 2 (*F*_1.59_ = 5.87, *p* = 0.019), but JHA did not affect the POPs of resident moths treated on the 3 to 4-day-old females (Figure 3A). Lifetime fecundity was also significantly affected by JHA treatment and the interaction between JHA treatment and age (Table 1). Exposure of both 1-day-old and 2-day-old resident females to JHA significantly produced more eggs than the controls (Day 1: *F*_1.58_ = 15.19, *p* < 0.001; Day 2: *F*_1.59_ = 4.75, *p* = 0.033), but the lifetime fecundity of those moths treated with JHA on the 3 to 4-day-old females was not significantly greater than the controls (Figure 3B). The oviposition period and male longevity were not affected by JHA treatment (Table 1, Figure 3C,G). Mating frequency and mating percentage differed significantly with JHA treatment (Table 1). The mating frequency of resident moths exposed to JHA on either day 1 or day 2 increased significantly compared to the controls (Day 1: *F*_1.58_ = 16.75, *p* < 0.001; Day 2: *F*_1.59_ = 4.35, *p* = 0.041, Figure 3D). However, the mating frequency was not affected by the JHA treatment on the 3-day-old and 4-day-old adults (Figure 3D) and 1-day-old resident females treated with JHA also had a greater mating percentage than the controls (*F*_1.58_ = 10.26, *p* = 0.002), but JHA did not affect the mating frequency of females treated on the 2 to 4-day-old (Figure 3E). JHA treatment, age and their interaction had significant effects on the female longevity (Table 1). Exposure of 1-day-old resident females to JHA significantly decreased their longevity compared to the controls (*F*_1.58_ = 33.84, *p* < 0.001), but the longevities of those moths by the JHA treatment on the 2 to 4-day-old females were not significantly shorter than the controls (Figure 3F). Only the JHA treatment significantly affected the ovarian development (Table 1). The ovarian development grade of resident females exposed to JHA on day 1 and day 2 after adult emergence was significantly higher than the controls (Day 1: *F*_1.58_ = 16.24, *p* < 0.001; Day 2: *F*_1.59_ = 4.96, *p* = 0.030, Figure 3H). However, it did not increase the ovarian development grade of resident females treated with JHA on the 3-day-old and 4-day-old (Figure 3H). In general, these results suggested that resident moths exposed to JHA on either day 1 or day 2 significantly accelerated the reproductive performance (Figure 3A–H).

In migrants, the POP decreased significantly for moths exposed to JHA on day 1 compared to the controls (*F*_1.58_ = 32.29, *p* < 0.001), but JHA did not affect the POPs of migrant females treated on those that were 2 to 4 days old (Figure 4A). Compared to the controls, 1-day-old migrant moths exposed to JHA significantly increased the lifetime fecundity (*F*_1.58_ = 13.42, *p* = 0.001, Figure 4B), mating frequency (*F*_1.58_ = 4.84, *p* = 0.032, Figure 4D) and mating percentage (*F*_1.58_ = 4.17, *p* = 0.046, Figure 4E). However, the lifetime fecundity, mating frequency and mating percentage of those moths by the JHA treatment on the 2 to 4-day-old females were not significantly higher than the controls (Figure 4B,D,E). There were no significant differences in the oviposition period and male longevity for JHA treatments compared with the controls at the same exposure age in the migrant groups (Figure 4C,G). Furthermore, 1-day-old migrant females exposed to JHA had a significantly shorter longevity compared to the controls (*F*_1.58_ = 24.32, *p* < 0.001, Figure 4F), but a greater ovarian development grade (*F*_1.58_ = 11.29, *p* = 0.001, Figure 4H). However, JHA did not affect the female longevity or ovarian development grade of migrant adults treated on those that were 2 to 4 days old (Figure 4F,H). The correlation analysis showed that JHA was positively correlated with reproduction of *M. separata* (Table 2). Taking the above results together, exposure to JHA significantly accelerated the reproduction of migrant females on the first day after adult eclosion (Figure 4A–G).

### 3.3. Effect of Age on Reproductive Performances of M. separata Populations Treated by JHA

The POP was also significantly affected by age (Table 1 and Table 3). In the JHA groups, 1-day-old moths had the shortest POP on all exposure ages, both in residents and migrants, which was significantly lower than those of day 2 to day 4 (Residents: *F*_3.113_ = 8.43, *p* < 0.001; Migrants: *F*_3.115_ = 5.12, *p* = 0.002, Table 3). However, no significant differences were found for the POP of residents and migrants in the control group (Table 3). Age had a significant effect on the lifetime fecundity of *M. separata* populations treated with JHA treatment (Table 1 and Table 3). The lifetime fecundity of 1-day-old females exposed to JHA was greater than those of 3-day-old and 4-day-old moths in residents (*F*_3.113_ = 3.59, *p* = 0.016), and that of 3-day-old moths in migrants (*F*_3.115_ = 2.92, *p* = 0.037), respectively (Table 3). However, no significant differences in lifetime fecundity of residents and migrants were found in the control groups (Table 3). Age did not significantly affect the oviposition period, mating frequency, mating percentage, male longevity and ovarian development grade of residents and migrants in both JHA and control groups (Table 1 and Table 3). The female longevity was significantly affected by age and the interaction between JHA treatment and age (Table 1 and Table 3). In residents, exposure of 1-day-old females to JHA significantly decreased their female longevity compared to the other exposure ages (*F*_3.113_ = 9.85, *p* < 0.001, Table 3). In migrants, the longevity of 1-day-old females exposed to JHA was also shorter than that of the 2-day-old moths (*F*_3.115_ = 3.01, *p* = 0.033), whereas it did not differ from the 3-day-old and 4-day-old moths. However, age did not affect the female longevity of residents and migrants in control groups (Table 3). The correlation analysis showed a negative tendency between age and reproduction of *M. separata* (Table 2).

## 4. Discussion

Our study showed that various reproductive traits exist between residents and migrants exposed to JHA in *M. separata*. Residents exhibited a shorter pre-oviposition period (POP) than the migrants in both JHA and control treatments, which was consistent with previous reports that residents showed stronger reproductive capacity [13,17,18,27]. Resident moths exposed to JHA on the first 2 days after adult emergence could facilitate their reproduction. By contrast, migrant moths exposed to JHA only on the first day after adult eclosion accelerated their reproductive performances.

Environment-dependent reproductive and flight capacity polyphenisms are important adaptive strategies that contribute to the ecological success of many insects [17,21,33,34]. Larvae of *M. separata* that experienced high larval density, poor nutrition and low temperature tend to develop into migrants [13,27,31]. The onset of migratory flight is generally initiated during the pre-oviposition stage for a variety of migratory noctuid species, which temporarily suppresses the ovarian development and mating behavior [25,35]. Migrants therefore invest more energy resources into the preparation for migration compared to residents, resulting in an increase in the pre-oviposition period and the reduction in lifetime fecundity [36]. Evidence of migratory period is usually associated with delay in oogenesis or prolonged POP, which has been reported in many insects [37,38]. Therefore, the prolonged pre-oviposition period is an important indicator to measure the migratory propensity of adult individuals [18]. Migrants of *M. separata* can be distinguished from residents by the longer pre-oviposition period, poorer reproductive capacity and stronger flight capacity [18,26], which is similar to *S. exigua* [11], *C. medinalis* [39] and *L. sticticalis* [12]. Migrants and residents displayed different reproductive traits, which perfectly matched the biological trade-off between migration and reproduction of the adult oriental armyworms.

In the oriental armyworm, there exists a short sensitive period for migrants, the first day after adult emergence [16,30]. In case of low temperature or starvation during the sensitive period, the temporary suppression of flight muscle development and the increased energy transfer from the flight system to the reproductive system occur in response to unfavorable natural conditions, which results in a shift from migrants to residents during the environmental sensitive period, by the method of accelerating the development of reproduction and decreasing the flight activity [30,31]. In the beet webworm, 24 h of starvation in the sensitive period (the first 2 days after adult emergence) of migrants may cause a transformation from migrants to residents [17]. Interestingly, *C**. medinalis* migrants in case of nutrient shortage on the first day after adult emergence would inhibit their reproduction, and the limited resources are invested in migration [15]. Although environmental stressors have different effects on reproductive traits of the adults during the sensitive period, it is the same that the short sensitivity periods for stimulus onset are only existent during the early stage of adults in some migratory insects [15,17,18]. Our results showed that the pre-oviposition periods of *M. separata* were significantly reduced for the residents on day 1 or day 2 and migrants on day 1 exposed to JHA. In contrast, exposure of residents and migrants on the other ages to JHA did not significantly influence the pre-oviposition period. Pearson’s correlation analysis also demonstrated that age had negative correlations with reproduction. These results indicated that the first 2 days after adult emergence might be the sensitive period for residents. In contrast, only the first day after adult emergence was the sensitive stage for migrants, which was consistent with previous reports that adult oriental armyworms could regulate their behavior of reproduction and migration during the first day after of adult emergence [16,30,40].

To our knowledge, juvenile hormone can regulate many physiological processes, such as reproduction, which is essential to the life cycle of insects [7,8,41]. It has been reported that ovarian development and maturation are controlled by JH, which regulates the synthesis and absorption of vitellogenin (Vg) and oocyte maturation in insects [42,43,44]. Our study substantiated that only 1-day-old migrants exposed to JHA significantly promoted their reproductive and ovarian development in *M. separata*, resulting in an increase in their lifetime fecundity, mating frequency, mating percentage and ovarian development grade, which was thus consistent with the results of previous work [27]. We further found that exposure of both 1-day-old and 2-day-old resident females to JHA significantly increased their lifetime fecundity, mating frequency and ovarian development grade. JHA significantly decreased the flight capacity and the flight muscle size of *M. separata* migrants on day 1 after adult emergence, but increased their reproduction, which played important roles in regulating the switch from migrants to residents during the early adult stage [16,27,30]. Thus, we presumed that residents and migrants of *M. separata* had different sensitive periods, which might be conducive to explaining why residents had a greater reproductive propensity but weaker flight capability than migrants [18].

## 5. Conclusions

In summary, our results showed that the reproductive traits of *M. separata* populations may be influenced by JHA and the days after adult emergence. The first two days and the first day after adult emergence may be the sensitive periods of *M. separata* residents and migrants, respectively. JHA significantly facilitated the reproduction of residents and migrants during their sensitive periods. These results confirmed the roles of JH in the physiological regulation of reproductive and ovarian development in *M. separata*. Moreover, our study is also beneficial to improve the accurate predictions of population dynamics and prevent outbreaks of *M. separata* in China.

## Figures and Tables

**Figure 1 insects-13-00506-f001:**
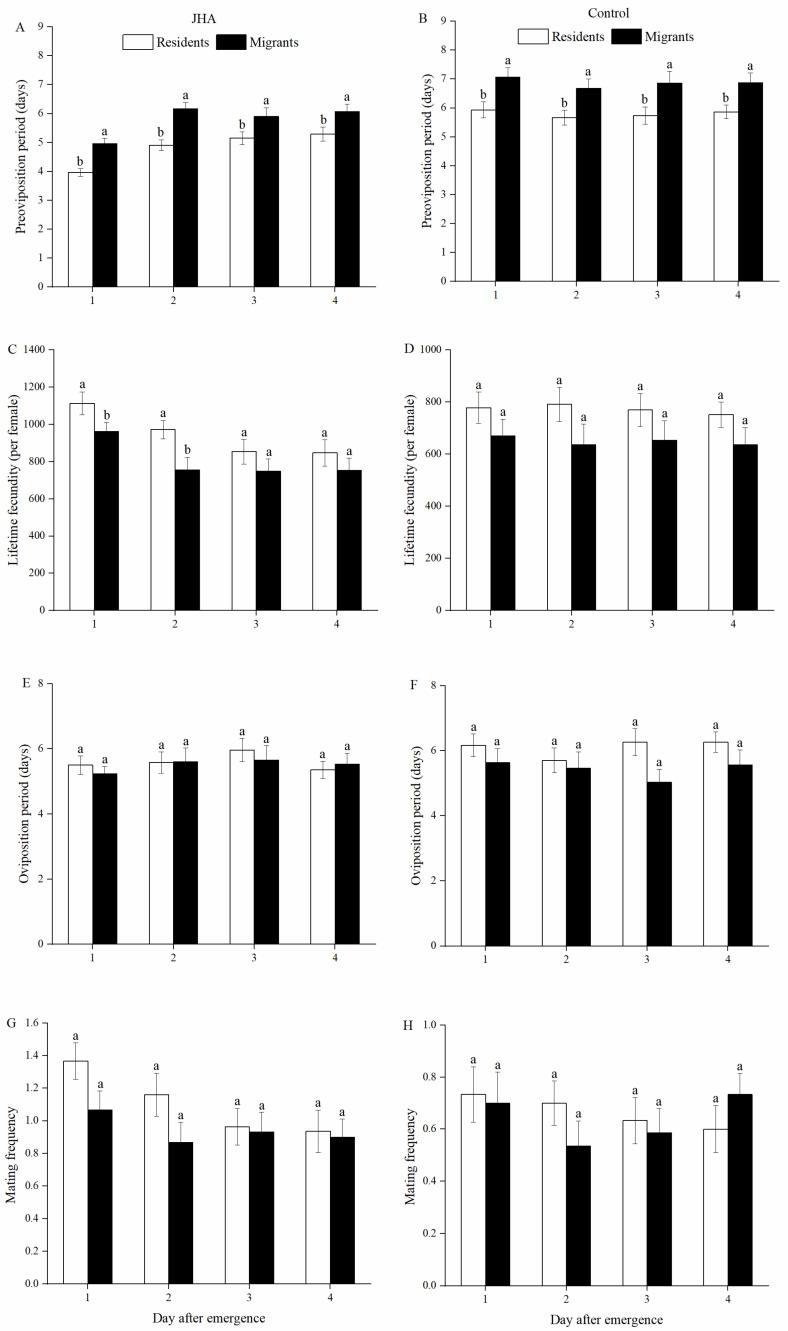
Pre-oviposition period (**A**: JHA, **B**: Control), lifetime fecundity (**C**: JHA, **D**: Control), oviposition period (**E**: JHA, **F**: Control) and mating frequency (**G**: JHA, **H**: Control) of *M. separata* between residents and migrants exposed to JHA at various ages. Different lowercase letters above group bars represent significant differences between residents and migrants at the 5% level by Tukey’s HSD test. Sample sizes for each JHA treatment (left to right), residents: 30, 31, 26 and 29 pairs; migrants: 30, 30, 29 and 30 pairs. Sample sizes for each control treatment (left to right), residents: 30, 30, 30 and 30 pairs; migrants: 30, 28, 29, and 30 pairs.

**Figure 2 insects-13-00506-f002:**
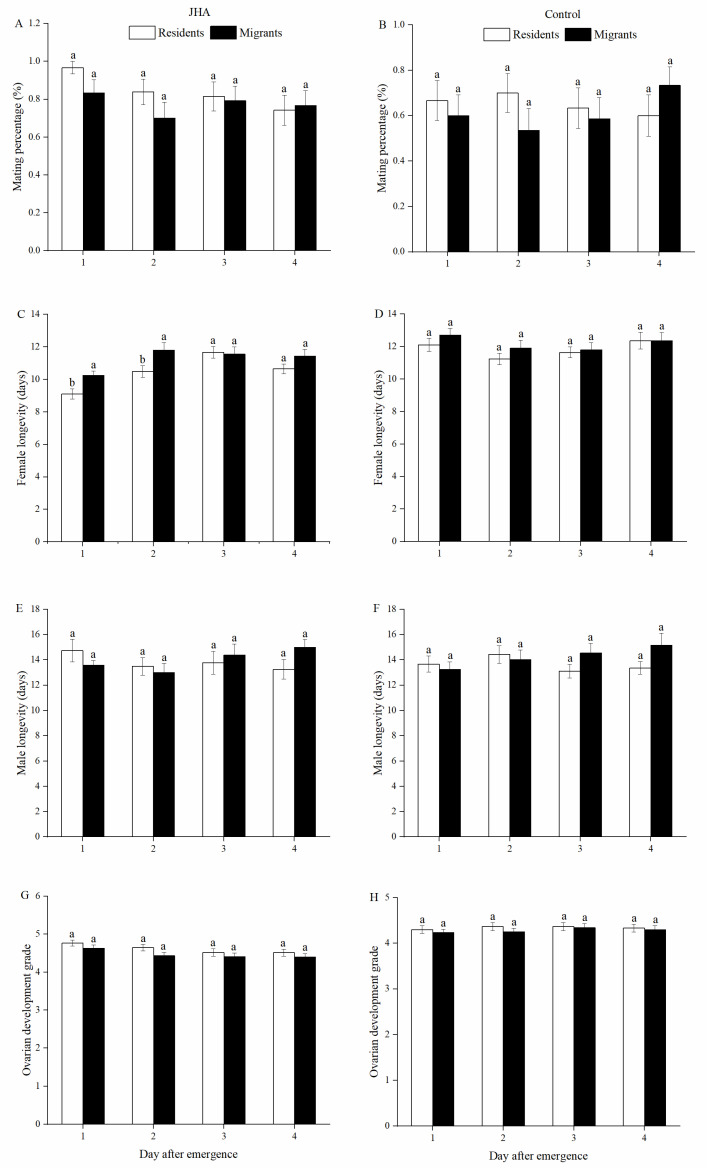
Mating percentage (**A**: JHA, **B**: Control), female longevity (**C**: JHA, **D**: Control), male longevity (**E**: JHA, **F**: Control) and ovarian development grade (**G**: JHA, **H**: Control) of *M. separata* between residents and migrants exposed to JHA at various ages. Different lowercase letters above group bars represent significant differences between residents and migrants at the 5% level by Tukey’s HSD test. Sample sizes for each JHA treatment (left to right), residents: 30, 31, 26 and 29 pairs; migrants: 30, 30, 29 and 30 pairs. Sample sizes for each control treatment (left to right), residents: 30, 30, 30 and 30 pairs; migrants: 30, 28, 29, and 30 pairs.

**Figure 3 insects-13-00506-f003:**
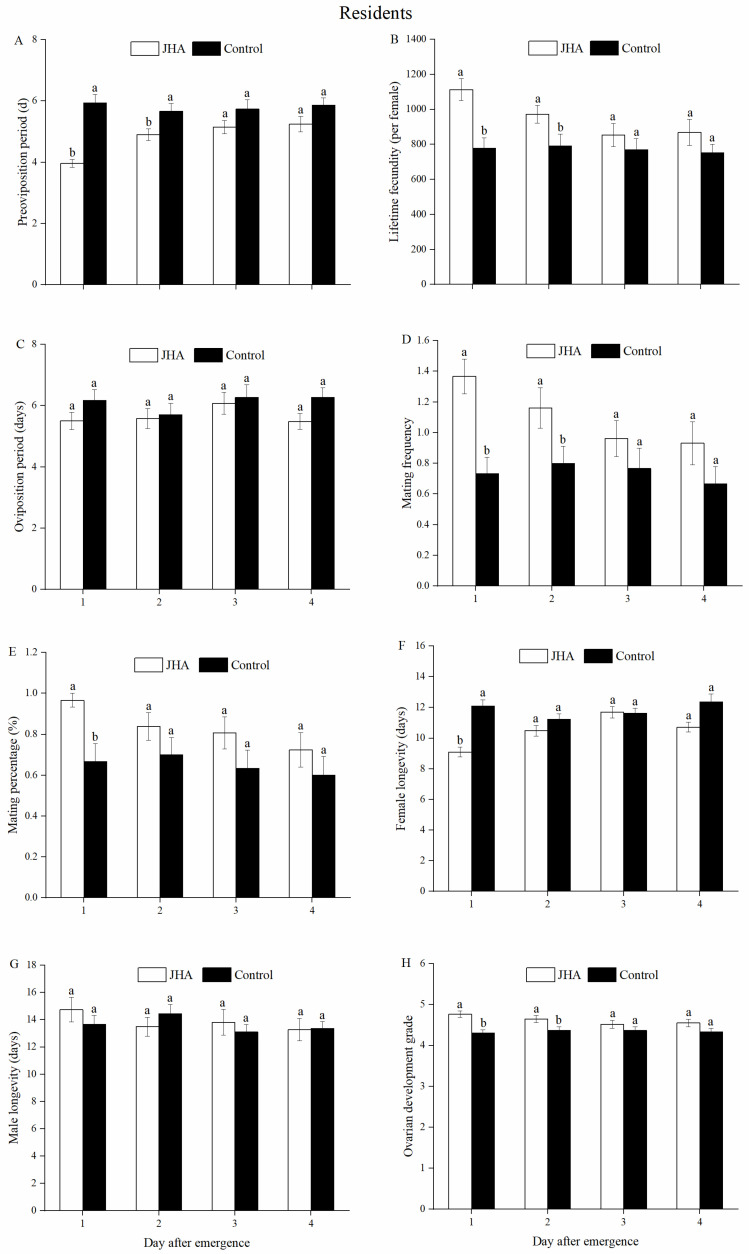
Reproductive performances of *M. separata* residents between the JHA and control groups at various ages. (**A**) Pre-oviposition period. (**B**) Lifetime fecundity. (**C**) Oviposition period. (**D**) Mating frequency. (**E**) Mating percentage. (**F**) Female longevity. (**G**) Male longevity. (**H**) Ovarian development grade. Different lowercase letters indicate significant differences between the JHA and control treatments at the 5% level by Tukey’s HSD test. Sample sizes (left to right) of JHA groups are 30, 31, 26 and 29 pairs, and of control groups are 30, 30, 30 and 30 pairs, respectively.

**Figure 4 insects-13-00506-f004:**
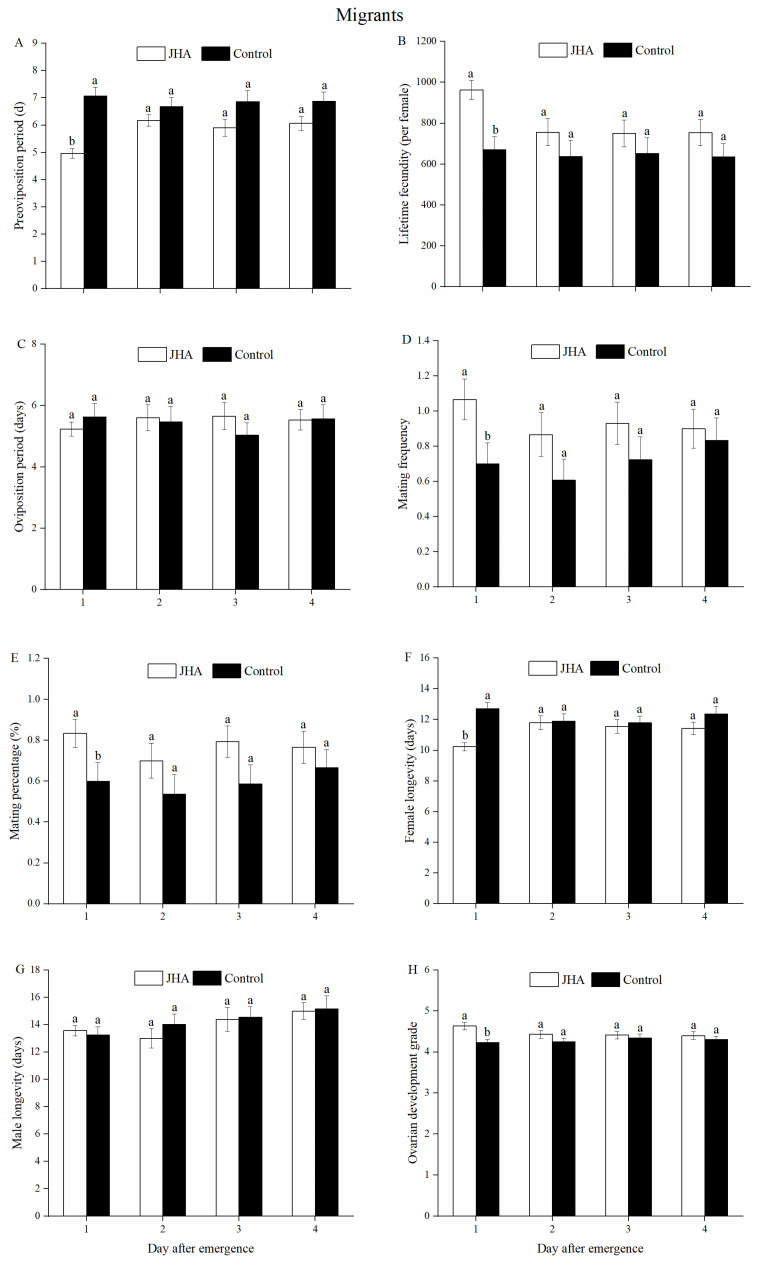
Reproductive performances of *M. separata* migrants between the JHA and control groups at various ages. (**A**) Pre-oviposition period. (**B**) Lifetime fecundity. (**C**) Oviposition period. (**D**) Mating frequency. (**E**) Mating percentage. (**F**) Female longevity. (**G**) Male longevity. (**H**) Ovarian development grade. Different lowercase letters indicate significant differences between the JHA and control treatments in the same bars at the 5% level by Tukey’s HSD test. Sample sizes (left to right) of JHA groups are 30, 30, 29 and 30 pairs, and of control groups are 30, 28, 29, and 30 pairs, respectively.

**Table 1 insects-13-00506-t001:** Summary of three-way ANOVA analysis on the effects of population, JHA treatment and age (days after emergence) on reproductive performances of *Mythimna separata*.

Reproductive Parameters	Source	*Df*	*F*	*p*
Preoviposition period (days)	Population	1	53.974	<0.001 *
	JHA treatment	1	53.619	<0.001 *
	Age	3	4.084	<0.001 *
	Population × JHA treatment	1	0.019	0.890
	Population × Age	3	0.638	0.591
	JHA treatment × Age	3	6.625	<0.001 *
	Population × JHA treatment × Age	3	0.179	0.836
	Sum of Squares Error	460		
Lifetime fecundity (per female)	Population	1	15.483	<0.001 *
	JHA treatment	1	28.635	<0.001 *
	Age	3	3.545	0.015 *
	Population × JHA treatment	1	0.145	0.703
	Population × Age	3	0.647	0.585
	JHA treatment × Age	3	2.628	0.050 *
	Population × JHA treatment × Age	3	0.156	0.856
	Sum of Squares Error	460		
Oviposition period (days)	Population	1	3.680	0.056
	JHA treatment	1	0.856	0.355
	Age	3	1.296	0.237
	Population × JHA treatment	1	1.091	0.297
	Population × Age	3	0.370	0.774
	JHA treatment × Age	3	0.545	0.652
	Population × JHA treatment × Age	3	0.196	0.822
	Sum of Squares Error	460		
Mating frequency	Population	1	1.616	0.204
	JHA treatment	1	18.532	<0.001 *
	Age	3	1.487	0.150
	Population × JHA treatment	1	1.755	0.186
	Population × Age	3	2.179	0.090
	JHA treatment × Age	3	1.945	0.121
	Population × JHA treatment × Age	3	0.204	0.815
	Sum of Squares Error	460		
Mating percentage (%)	Population	1	1.365	0.243
	JHA treatment	1	14.986	<0.001 *
	Age	3	1.406	0.183
	Population × JHA treatment	1	0.026	0.871
	Population × Age	3	1.817	0.143
	JHA treatment × Age	3	0.726	0.537
	Population × JHA treatment × Age	3	0.177	0.838
	Sum of Squares Error	460		
Female longevity (days)	Population	1	10.454	0.001 *
	JHA treatment	1	24.492	<0.001 *
	Age	3	2.637	0.006 *
	Population × JHA treatment	1	2.573	0.109
	Population × Age	3	0.618	0.603
	JHA treatment × Age	3	8.663	<0.001 *
	Population × JHA treatment × Age	3	0.090	0.914
	Sum of Squares Error	460		
Male longevity (days)	Population	1	1.060	0.304
	JHA treatment	1	0.003	0.957
	Age	3	0.716	0.694
	Population × JHA treatment	1	0.264	0.607
	Population × Age	3	2.778	0.041 *
	JHA treatment × Age	3	1.160	0.325
	Population × JHA treatment × Age	3	0.213	0.808
	Sum of Squares Error	460		
Ovarian development grade	Population	1	1.814	0.064
	JHA treatment	1	27.819	<0.001 *
	Age	3	0.919	0.508
	Population × JHA treatment	1	0.813	0.368
	Population × Age	3	0.250	0.861
	JHA treatment × Age	3	2.451	0.063
	Population × JHA treatment × Age	3	0.004	0.996
	Sum of Squares Error	460		

* Significant *p* value.

**Table 2 insects-13-00506-t002:** Pearson’s correlations of reproduction of *M. separata* with populations, JHA treatment and age.

Parameters	Population		JHA Treatment		Age (Days after Emergence)	
	*r*	*p*-Value	*r*	*p*-Value	*r*	*p*-Value
Preoviposition period	0.305	<0.001 *	−0.318	<0.001 *	0.148	0.001 *
Lifetime fecundity	−0.185	<0.001 *	0.237	<0.001 *	−0.127	0.006 *
Oviposition period	−0.097	0.035 *	−0.043	0.346	0.009	0.839
Mating frequency	−0.060	0.196	0.203	<0.001 *	−0.059	0.198
Mating percentage	−0.057	0.219	0.186	<0.001 *	−0.022	0.629
Female longevity	0.120	0.009 *	−0.245	<0.001 *	0.129	0.005 *
Male longevity	0.048	0.297	0.000	0.992	0.055	0.231
Ovarian development grade	−0.107	0.020 *	0.239	<0.001 *	−0.066	0.154

* Significant *p* value.

**Table 3 insects-13-00506-t003:** Effect of age on reproductive performances of *M. separata* populations treated by JHA.

Parameters	Population	JHA Treatment	Age (Days after Emergence)			
			1	2	3	4
Preoviposition period (days)	Residents	JHA	3.97 ± 0.13 b	4.90 ± 0.19 a	5.08 ± 0.21 a	5.24 ± 0.25 a
		Control	5.93 ± 0.28 a	5.67 ± 0.25 a	5.73 ± 0.30 a	5.87 ± 0.23 a
	Migrants	JHA	4.97 ± 0.18 b	6.17 ± 0.21 a	5.89 ± 0.31 a	6.07 ± 0.26 a
		Control	7.07 ± 0.32 a	6.68 ± 0.33 a	6.86 ± 0.40 a	6.87 ± 0.34 a
Lifetime fecundity (per female)	Residents	JHA	1125.52 ± 62.56 a	971.90 ± 50.04 ab	879.62 ± 63.43 b	867.93 ± 74.02 b
		Control	778.43 ± 59.67 a	791.27 ± 66.55 a	769.40 ± 63.69 a	751.43 ± 48.26 a
	Migrants	JHA	962.40 ± 47.12 a	756.17 ± 66.55 ab	749.45 ± 65.45 b	754.03 ± 64.56 ab
		Control	670.40 ± 64.29 a	636.71 ± 78.41 a	653.03 ± 75.56 a	636.07 ± 65.49 a
Oviposition period (days)	Residents	JHA	5.50 ± 0.28 a	5.58 ± 0.32 a	6.08 ± 0.35 a	5.48 ± 0.26 a
		Control	6.17 ± 0.35 a	5.70 ± 0.37 a	6.27 ± 0.31 a	6.21 ± 0.31 a
	Migrants	JHA	5.23 ± 0.23 a	5.60 ± 0.43 a	5.66 ± 0.44 a	5.53 ± 0.33 a
		Control	5.53 ± 0.43 a	5.46 ± 0.50 a	5.03 ± 0.40 a	5.57 ± 0.46 a
Mating frequency	Residents	JHA	1.37 ± 0.11 a	1.16 ± 0.13 a	0.96 ± 0.12 a	0.93 ±0.14 a
		Control	0.73 ± 0.11 a	0.80 ± 0.11 a	0.77 ± 0.13 a	0.67 ± 0.11 a
	Migrants	JHA	1.07 ± 0.12 a	0.87 ± 0.12 a	0.93 ± 0.12 a	0.90 ± 0.11 a
		Control	0.70 ± 0.12 a	0.61 ± 0.12 a	0.72 ± 0.13 a	0.83 ± 0.13 a
Mating percentage (%)	Residents	JHA	96.67 ± 3.33 a	83.87 ± 6.72 a	80.77 ± 7.88 a	72.41 ± 8.45 a
		Control	66.67 ± 8.75 a	70.00 ± 8.23 a	63.33± 8.95 a	60.00 ± 4.44 a
	Migrants	JHA	83.33 ± 6.92 a	70.00 ± 8.51 a	79.31 ± 7.66 a	76.67 ± 7.85 a
		Control	60.00 ± 9.10 a	53.57 ± 9.60 a	58.62 ± 9.31 a	66.67 ± 8.75 a
Female longevity (days)	Residents	JHA	9.10 ± 0.32 b	10.48 ± 0.35 a	11.63 ± 0.33 a	10.72 ± 0.32 a
		Control	12.10 ± 0.41 a	11.23 ± 0.35 a	11.69 ± 0.36 a	11.36 ± 0.38 a
	Migrants	JHA	10.23 ± 0.27 b	11.80 ± 0.46 a	11.55 ± 0.45 ab	11.43 ± 0.41 ab
		Control	12.70 ± 0.42 a	11.89 ± 0.50 a	11.79 ± 0.44 a	12.37 ± 0.51 a
Male longevity (days)	Residents	JHA	14.73 ± 0.90 a	13.48 ± 0.70 a	13.80 ± 0.94 a	13.28 ± 0.82 a
		Control	13.67 ± 0.64 a	14.43 ± 0.67 a	13.10 ± 0.54 a	13.37 ± 0.49 a
	Migrants	JHA	13.57 ± 0.39 a	13.00 ± 0.72 a	14.38 ± 0.88 a	15.00 ± 0.41 a
		Control	13.23 ± 0.61 a	14.04 ± 0.75 a	14.55 ± 0.76 a	15.17 ± 0.94 a
Ovarian development grade	Residents	JHA	4.77 ± 0.08 a	4.65 ± 0.09 a	4.54 ± 0.10 a	4.55 ± 0.09 a
		Control	4.30 ± 0.09 a	4.37 ± 0.09 a	4.37 ± 0.09 a	4.33 ± 0.09 a
	Migrants	JHA	4.63 ± 0.09 a	4.43 ± 0.09 a	4.41 ± 0.09 a	4.40 ± 0.09 a
		Control	4.23 ± 0.08 a	4.25 ± 0.08 a	4.34 ± 0.09 a	4.30 ± 0.09 a

Data in the table are mean ± SE. Different lowercase letters in the same row represent significant differences among different ages by Tukey’s HSD test at the 5% level. The sample sizes for each JHA treatment from residents are 30, 31, 26 and 29 pairs, and each control treatment from residents are 30, 30, 30 and 30 pairs, from left to right, respectively. Sample sizes (left to right) of JHA groups from migrants: 30, 30, 29 and 30 pairs, and of control groups from migrants: 30, 28, 29, and 30 pairs.

## Data Availability

The data presented in this study are available on request from the corresponding author.

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
