# Peer review of "Effects of Juvenile Hormone Analog and Days after Emergence on the Reproduction of Oriental Armyworm, Mythimna separata (Lepidoptera: Noctuidae) Populations"

_insects, 2022, doi:10.3390/insects13060506_

Round 1

Reviewer 1 Report

There are several things the authors must clarify before the study is published.  Given all of their previous work on the effects of JH on reproductive responses of migrants and residents of this same species, this study is based on the premise that additional subtle differences exist and authors state that more emphasis is needed on resident populations.  This necessarily requires that well-characterized and stable experimental populations of resident vs. migrant insects be utilized. In other words, it must be assured that migrant vs. resident colonies are consistently maintaining their status. Yet the authors also know and point out that environmental variables can markedly influence conversion from one form to another.  The methods give scant attention to this important aspect and do not provide the necessary certainty that migrant vs. resident colonies have the required reliability in their physiological and behavioral status.  Authors should explain and clarify how colonies meet these requirements and how migratory status is known and clearly distinct between the two colonies.

Also in methodology, were treated and untreated insects paired for mating? Or were pairs always chosen from the same treatment group? Clarify.

What does treated "at 15 pm every day" mean?  15:00 hours?

An important aspect of methodology that bears directly on statistical testing is whether the same insects were treated cumulatively on day 1, 2, 3 and 4, or were separate individuals treated at each day and only on that day?  If the former, a Repeated Measures Anova should be used to test the age factor since the insects used at each day are not independent of the previous day. 

In Results, abbreviations such as POP should be defined at the first mention.  More importantly, the initial presentation of results is confusing.  The table of means (Table 2) is not easy to read due to the use of two rows for the name of each response variable, when actually the two rows correspond to treatment vs. control.  Ideally, tables with means together with statistical summary should be integrated together.

In terms of methodology, the authors place much importance on determining the effects of JH on migrant vs. resident insects.  This implies that resident/migrant status should be a factor in the experimental design and in statistical testing, which it is not. Resident and migratory insects were  analyzed separately as separate experiments.  Including migratory status as a factor with its two levels would allow not only testing of main effects of migratory status but importantly its interactions with age and JH treatment.

Author Response

Dear Reviewer,

        Thanks very much for reviewers' suggestions on our manuscript entitled “Effects of juvenile hormone analog and age on the reproduction of oriental armyworm, Mythimna separata (Lepidoptera: Noctuidae) populations” (manuscript # insects-1721806). We have read these comments carefully and made major modification correspondingly. We have labeled all the modifications we have made in pink and also the answers to these questions including line numbers. In addition, changes in the revised manuscript were highlighted using red font.

Once again, we acknowledge your comments very much, which are valuable and meaningful in improving the quality of our manuscript.

Sincerely yours.

Weixiang Lyu

2022-5-17

Reviewer 2 Report

Lv et al., demonstrate the effects of JHA on the reproduction of the oriental armyworm. In addition, the authors compared the effects of JHA on two different colonies, migrants and residents of this species. Also, the effects were analyzed in the adult moths at different timing.

However, it is very difficult to understand the figures. The authors should explain more about the results.

Authors should address major points and minor points as listed below.

Major)

1) The term “Age” should be changed. For example, post-eclosion duration or something. Although reference 6 uses “Age”, Lorenz uses this term as a longer period.

2) There was no comparison data between migrants and residents. The authors should address and explain this comparison.

3) there were no descriptions of the number of the examined individuals. Authors must indicate the numbers. Also, authors must indicate the reproducibility of the experiments.

4) There are no explanations for the relationship between reproduction and several events examined in this study. Authors must explain the design of the experiments.

Minor)

Simple Summary;

L8; JHA: this JHA is the first appearance.

Abstract:

L2: However: This “However” is not needed.

L5 and later; JHA titer: Authors used “titer”. But titer means different from indications by authors. Also, there are no data for the titer, using different concentrations for testing the effects.

Introduction;

L3: oogenesis-flight syndrome: authors must explain this unfamiliar term.

4th para, L3: there were no explanations in this manuscript. Please explain in more detail. For example, a high or low level of JH causes some phenotype.

L11: the sentence beginning with “However, little is known about …”. This sentence is not needed.

L17: The sentences from “This study expands out …” and “Moreover, our study is also…” are not needed because these sentences are duplicated in conclusion.

Materials and Methods;

2.1; Colonies; in this manuscript, the authors use “populations” and “colonies”. Are they identical meanings?

2.2: L2: with a JHA, Methoprene (Sigma Chemical …??

2.4:L12: “mean”? Authors should use the more appropriate word.

Day-1 is the first day after eclosion?? Or Day-0?? Please explain in this session.

Results;

Some sentences did not show the results although the figures indicated. Authors should re-write thoroughly carefully.

Table 1; there are no explanations (d), error, … Please re-write the table.

Table 2; There are no explanations aA, and others…

Figure 1; No explanations for the numbers of individuals examined.

Figure 2; The vertical axis explanations are different from the legends in the figure. No description for the number of individuals examined.

Figure legends; several grammatical errors were found.

Discussion;

L1: “differentiation”; authors should use an appropriate word.

The contents of references 28, 29 should explain in the Introduction briefly. Readers may want to understand more about the background of this study.

Author Response

Dear Reviewer,

        Thanks very much for reviewers' suggestions on our manuscript entitled “Effects of juvenile hormone analog and age on the reproduction of oriental armyworm, Mythimna separata (Lepidoptera: Noctuidae) populations” (manuscript # insects-1721806). We have read these comments carefully and made major modification correspondingly. We have labeled all the modifications we have made in pink and also the answers to these questions including line numbers. In addition, changes in the revised manuscript were highlighted using red font.

Once again, we acknowledge your comments very much, which are valuable and meaningful in improving the quality of our manuscript.

Sincerely yours.

Reviewer 3 Report

The manuscript submitted by Lv and collaborators presents a series of experiments aimed to characterize the effect of an insecticide analog of insect’s juvenile hormone, on two strains of a pest moth. The information provided is interesting and the work is rich in results.

There are, however, some points requesting the attention of the authors:

  • It is quite surprising that the authors made the choice of presenting methoprene as just a JH analog, rather than as it is commercialized and mostly employed, i.e. an insecticide. This aspect is not mentioned at all and it is particularly curious in a study concerning an agricultural pest.
  • Title: in line with the previous point, replacing “…juvenile hormone analog…” by “…methoprene…” seems appropriate. This is just a suggestion, not mandatory.
  • Keywords: All the terms chosen by the authors are already in the title. The role of keywords is to complete the title, providing additional information for automatic searching. Please, replace them with words not mentioned in the title.
  • Methods: The origins and age of the strains must be provided and the choice of the dose and application method of methoprene justified. Does it correspond to the insecticide recommended dose? Is this pharmacological? Is this amount employed with other insects for studying JH action?
  • Statistics: A parametric test, i.e. ANOVA, has been used for the statistical analysis of the results. Yet, no indication about the distribution of the data. Before applying it, the normality, homoscedasticity, and symmetry of data distribution must be verified and, if these tests are not passed, either non-parametric test should be applied or data transformed. No indication about these verifications is provided. Indeed, this reviewer has the impression that they have not been conducted because they apparently applied ANOVA for testing percentages, which by definition do not follow a normal distribution. Please, verify and correct if necessary.
  • Results: It is hard for the reader to follow the many tables, figures, and data in the text. My advice would be to organize the results hierarchically, supported by figures and presenting tables as supplementary material.
  • Authors use the expression “JHA titer” for referring to treated or untreated groups. Yet, no “titer” has been measured or provided. Please, change for proper terms.
  • In some passages, the authors underline that their findings are important for understanding complex physiological processes and enriching control strategies. These two statements need to be properly developed, adequately explaining how and why they think that this could be the case. Otherwise, they appear as overselling arguments.

Author Response

(The authors gave the same response as above.)

Round 2

Reviewer 1 Report

I can now support publication of this study.  The authors have addressed my major concerns.  Some minor copy editing may still be required.

Author Response

Dear reviewer:

Thanks very much for reviewers' suggestions on our manuscript entitled “Effects of juvenile hormone analog and age on the reproduction of oriental armyworm, Mythimna separata (Lepidoptera: Noctuidae) populations” (manuscript # insects-1721806). We have read your comments carefully and made the modification correspondingly. We have labeled all the modifications we have made in pink and also the answers to these questions including line numbers. In addition, changes in the revised manuscript were highlighted using red font.

Once again, we acknowledge your comments very much, which are valuable and meaningful in improving the quality of our manuscript.

Sincerely yours.

Weixiang Lyu

2022-5-24

Reviewer 2 Report

The authors have carefully revised most of the points according to the previous round of review. As for the term “Age” in the title, the authors might confuse the context of this manuscript as the senescence in the adult periods. Then, please carefully use the term “age” throughout this manuscript. As this reviewer suggested in the previous round, post-eclosion or post-adult emergence or similar phrases should be used.

This reviewer has confirmed the points that the authors have revised.

Response 1: Thanks for your comments. We have adopted three-way analysis of variance with population (Residents or Migrants), JHA treatment (JHA or acetone) and exposure age after adult emergence (including 1, 2, 3 and 4 days old) as factors to assess the effects of JHA and age on reproduction of M. separata populations according to the reviewers' suggestions. (Line 250-252).

We have re-written the results according to your advice. The results were hierarchically organized from 3 parts (namely, population, JHA treatment and age), including all the tables, figures and data in the revision (Line 173-290).

à OK

Response 2: Thanks for your suggestion. We have added some interpretations of age “Adult age in this experiment refers to the days after emergence. The first day of adult emergence is considered day 1, the next day is day 2, and so forth in the revision (Line 135-136).

à The title should be thought again. Post-eclosion aging might be fine.

Response 3: Thanks for your suggestion. We have added the comparison data between migrants and residents, and provided with the explanations in the results according to your suggestion (Line 173-194).

à OK

Response 4: Thanks for your suggestion. We have added the reproducibility of the experiments in the revision according to your suggestion (Line132-133, 141-142). We have also added the number of the examined individuals for each treatment in the notes of Table 2, Figure 1, Figure 2, Figure 3 and Figure 4 in the revision (Line 254-259, 263-270, 272-279, 281-284, 286-290).

à OK

Response 5: Thanks for your suggestion. We have added the correlation analysis between reproduction indicators and other factors including population, JHA treatment and age in Results. (Line 193-194, Line 229-230, Line 246-248,  Line 260-261, Table 3).

à OK

Response 6: Thanks for your suggestion. We have changed “JHA” to “JH analog (JHA)” in the revision (Line 21).

à OK

Response 7: Thanks for your suggestion. We have deleted the word “However” in the revision (Line 26).

à OK

Response 8: Thanks for your suggestion. We have deleted the word “titer” throughout the revision (Line 21, 29, 56, 93, 99 and so on). Actually, the adults were treated with either 5 μL JHA (6 μg/μL) or the same volume of acetone solution as control (Line 126-129).

àOK

Response 9: Thanks for your suggestion. We have added some explanations “In migratory insects, the onset of migratory behaviour usually occurs in the pre-reproduction period, which is initiated by sexually immature adults.” to further explain the term “oogenesis-flight syndrome” in the revision (Line 45-47).

à OK

Response 10: Thanks for your suggestion. We have added some explanations “Levels of JH in the residents were significantly greater than those in the migrants on the first to sixth day after emergence (Zhang et al., 2020). Jiang and Luo also suggested that lower JH levels favored adult migration, whereas higher levels stimulated oogenesis (Jiang et al., 2005). in the revision (Line 90-93).  

àOK

Response 11: Thanks for your suggestion. We have deleted this sentence in the revision (Line 99).

àOK

Response 12: Thanks for your suggestion. We have deleted this sentence in the revision (Line 103).

à OK

Response 13: Thanks a lot for your questions. We have changed “colonies” to “populations” throughout   the revision (Line 107,116, 123).

àOK

Response 14: Thanks for your suggestion. We have changed “with juvenile hormone analogue called methoprene” to “with a juvenile hormone analogue, Methoprene” in the revision (Line 122-125).

àOK

Response 15: Thanks for your suggestion. We have changed “mean” to “represented” in the revision (Line 155).

à OK

Response 16: Thanks for your question. We have added some explanations “Adult age in this experiment refers to the days after emergence. The first day of adult emergence is considered day 1, the next day is day 2, and so forth” in the revision (Line 135-136).

àOK

Response 17: Thanks for your suggestion. We have re-written the results according to your advice. The results were hierarchically organized from 3 parts (namely, population, JHA treatment and age), including all the tables, figures and data in the revision (Line 172-290).

Point 18: Results, Table 1: there are no explanations (d), error, … Please re-write the table.

Response 18: Thanks for your suggestion. We have changed “(d)” to “(days)”, and the word “error” refers to “Sum of Squares Error” in Table 1. We have also re-written Table 1 to investigate the effects of population, JHA treatment and age on reproductive performances of Mythimna separata by three-way ANOVA analysis in the revision (Line 250-252).

Point 19: Results, Table 2: There are no explanations aA, and others…

Response 19: Thanks for your suggestion. We have re-written Table 2 in the revision. Different lowercase letters in the same row represent significant differences among different ages by Tukey’s HSD test at 5% level (Line 254-259).

àOK

Response 20: Thanks for your suggestion. We have added the number of the examined individuals for each treatment in the notes of Figure 1, Figure 2, Figure 3 and Figure 4 in the revision (Line 263-270, 272-279, 281-284, 286-290).

àOK

Response 21: Thanks for your suggestion. We have re-drawn Figure 1, Figure 2, Figure 3 and Figure 4 in our manuscript. We have added the numbers of individuals examined in the notes of each figure. The grammatical errors found in the Figure legends have also been revised(Line 260-263, 272-275, 281-284, 288-290, 294-296).

àOK

Response 22: Thanks your suggestion. We have changed “differentiation” to “variation” in the revision (Line 292).

à OK

Response 23: Thanks for your suggestion. We have changed two sentences to “During this sensitive period of the first day after adult emergence, JH may regulate the shift from migrants to residents in M. separata [27, 30].” in the revision (Line 95-96).

àOK

Author Response

Dear reviewer:

Thanks very much for your' suggestions on our manuscript entitled “Effects of juvenile hormone analog and age on the reproduction of oriental armyworm, Mythimna separata (Lepidoptera: Noctuidae) populations” (manuscript # insects-1721806). We have read your comments carefully and made the modification correspondingly. We have labeled all the modifications we have made in pink and also the answers to these questions including line numbers. In addition, changes in the revised manuscript were highlighted using red font.

Once again, we acknowledge your comments very much, which are valuable and meaningful in improving the quality of our manuscript.

Sincerely yours.

Weixiang Lyu

2022-5-24

Reviewer 3 Report

Thank you very much for your responses and the improvements introduced in your manuscript.

Author Response

Dear reviewer:

Thanks very much for your suggestions on our manuscript entitled “Effects of juvenile hormone analog and age on the reproduction of oriental armyworm, Mythimna separata (Lepidoptera: Noctuidae) populations” (manuscript # insects-1721806). We have read your comments carefully and made the modification correspondingly. We have labeled all the modifications we have made in pink and also the answers to these questions including line numbers. In addition, changes in the revised manuscript were highlighted using red font.

Once again, we acknowledge your comments very much, which are valuable and meaningful in improving the quality of our manuscript.

Sincerely yours.

Weixiang Lyu

2022-5-24
